# Can C-Band SAR be used to estimate soil organic carbon storage in tundra?

Annett Bartsch[1,2], Barbara Widhalm[1,2], Peter Kuhry[3], Gustaf Hugelius[3], Juri Palmtag[3], and Matthias Benjamin Siewert[3]

[1]Zentralanstalt für Meteorologie und Geodynamik, 1190 Vienna, Austria
[2]Vienna University of Technology, 1040 Vienna, Austria
[3]Stockholm University, Department of Physical Geography, SE-106 91 Stockholm, Sweden

*Correspondence to:* Annett Bartsch (annett.bartsch@zamg.ac.at)

**Abstract.** A new approach for the estimation of Soil Organic Carbon (SOC) pools North of the tree line has been developed based on synthetic aperture radar data (SAR; ENVISAT Advanced SAR Global Monitoring mode). SOC values are directly determined from backscatter values instead of up-scaling using land cover or soil classes. The multi-mode capability of SAR allows application across scales. It can be shown that measurements in C-band under frozen conditions represent vegetation and surface structure properties which relate to soil properties, specifically SOC. It is estimated that at least 29 PgC are stored in the upper 30 cm of soils North of the tree line. This is approximately 25% less than stocks derived from the soil map based Northern Circumpolar Soil Carbon Database (NCSCD). The total stored carbon is underestimated since the established empirical relationship is not valid for peatlands as well as strongly cryoturbated soils. The approach does however provide the first spatially consistent account of soil organic carbon across the Arctic. Furthermore, it could be shown that values obtained from 1 km resolution SAR correspond to accounts based on a high spatial resolution (2 m) land cover map over a study area of about 7 x 7 km in NE Siberia. The approach can be also potentially transferred to medium resolution C-band SAR data such as ENVISAT ASAR Wide Swath with 120 m resolution but it is in general limited to regions without woody vegetation. Global Monitoring Mode derived SOC increases with unfrozen period length. This indicates the importance of this parameter for modelling of the spatial distribution of soil organic carbon storage.

# 1 Introduction

The quantification of presently stored soil organic carbon (SOC) in the Arctic is of high interest for the assessment of climate change impacts in this environment (Schuur et al., 2008). These carbon pools are prone to changes, specifically increasing temperatures which are predicted for large proportions of the Arctic. Degradation of the underlying permafrost may induce environmental changes that trigger or accelerate the release of greenhouse gases at a scale that its impact is expected to be relevant to the global climate (Schuur et al., 2015).

The Northern Circumpolar Soil Carbon Database (NCSCD) by Tarnocai et al. (2009) and recently updated in Hugelius et al. (2013) provides currently the only basis for circumpolar accounts of soil organic carbon storage. Accounts for carbon stored in soils down to 3 m depth and additional stocks of sediments with various thicknesses are included. It relies on regionally differing information sources, including soil maps. This leads to differences in accuracies across the Arctic and up-scaling artifacts. Uncertainties in the SOC estimates for the northern permafrost region are large (Tarnocai et al., 2009).

SOC can be derived from remotely sensed data using soil color as an indicator (Wulf et al., 2015). The assumption is that SOC is related to wetness which influences the soil color. But this approach is only applicable in cases without vegetation cover. A further method is the combination of in situ measurements with land cover maps. Soil carbon and nitrogen upscaling down to a depth of 100 cm based on land cover maps has been shown feasible on site scale (Hugelius et al., 2011; Palmtag et al., 2015; Siewert et al., 2015). Detailed, site specific land form and/or land cover classification schemes using high resolution satellite data have been applied. The method can be used to provide a weighed 'landscape-level' mean for the entire study area, for which a single SOC value is attributed to each recognized thematic class. Spatially explicit variations within a certain class and transitions cannot be derived.

The major constrain for upscaling to circumpolar scale with such a method is the insufficient thematic detail of existing land cover datasets (e.g. Widhalm et al. (2015)). An approach which makes use of satellite data available at multiple scales is required for the upscaling to larger regions and up to circumpolar levels. One option would be the development of a land cover dataset which includes the required thematic detail in order to represent the range of carbon stocks across the high latitudes. A geospatial product which has been proven applicable for many studies is the Circumpolar Arctic Vegetation Map (CAVM) by Walker et al. (2002). It shows the types of vegetation that occur across the Arctic, between the ice-covered Arctic Ocean to the north and the northern limit of forests to the south. It is, however, designed to map vegetation communities rather than soil types. Ping et al. (2008) used the CAVM to derive four distinct landscape units in order to upscale SOC for Northern America North of the treeline. A wide range of SOC stock values have been observed for landscape unit averages due to environmental gradient effects. A direct relationship between the Normalized Difference Vegetation Index (NDVI) and SOC has been found for the high arctic (Horwath Burnham and Sletten, 2010) but has not been shown applicable outside that region. An alternative approach for direct derivation of SOC values is required which can represent gradients and is applicable for the entire arctic and subarctic domain.

The land cover types which are of interest differ in surface structure, including vascular plant cover as well as in micro-topographic relief (e.g. tussocks and hummocks). Such features can be captured with active microwave data depending on

frequency and polarization. The signal interacts with these surfaces and a certain proportion is directed back to the sensor. The backscatter intensity can be thus used to obtain information about the surface properties. Radar satellite data are available on different scales (meters to kilometers) based on usage of the Synthetic Aperture Radar (SAR) principle. A limitation of high to medium resolution applications is the variable coverage (Bartsch et al., 2009, 2012). ENVISAT ASAR data acquired in Global

Monitoring Mode are however available circumpolar spanning several years (Bartsch et al., 2009; Widhalm et al., 2015) with approximately 1 km resolution. An additional advantage of such data is the illumination independence. A challenge is the complexity of the interaction of the signal with the earth surface. Water content of the near surface soil contributes to the backscatter during unfrozen periods (Wagner et al., 1999; Pathe et al., 2009) as well as snow grains in the winter time (Ulaby et al., 1982). The impact of the latter is however limited during early winter when snow cover is low and metamorphosis of snow grains

negligible at C-band (Naeimi et al., 2012). Such data are not only available from SAR, but also from the much coarser spatial resolution scatterometer data. Across scale and instrument applications are common. ENVISAT ASAR data acquired in Wide Swath (WS, ~120 m) and Global Monitoring (GM, ~1 km) mode were e.g. used to downscale soil moisture related patterns (Wagner et al., 2008; Pathe et al., 2009) or bias correct (Högström and Bartsch, accepted) information from scatterometer. The applicability of similar multitemporal analyses for both GM and WS data has been already demonstrated for forest growing

stock volume retrieval (Santoro et al., 2011). WS data resampled to GM resolution is also suggested as substitute in case of missing GM records. In this study it is hypothesized that a relationship exists between SOC and the C-band radar backscatter resulting from surface roughness in tundra regions which is valid for SAR data acquired at different spatial resolutions. The aim is to provide a circumpolar consistent account of SOC which also provides information on gradients. Field measurements of SOC and land cover map based up-scaling results are used and results are cross-compared to externally available soil type

information (NCSCD and in situ) as well as satellite derived and potentially related parameters (vegetation and growing season length).

## 2 Datasets

### 2.1 Synthetic aperture radar data

The ASAR (Advanced Synthetic Aperture Radar) instrument on-board ESA's ENVISAT satellite operated in C-Band (5.3

GHz) in five different modes with varying temporal and spatial resolution from 30 m to 1 km from 2002 to spring 2012. Among these, the Global Monitoring Mode (GM) formed the background mission and was active whenever no other mode had been requested. GM data were obtained using the ScanSAR technique and provided low resolution images (1 km) with a wide swath width of 405 km and incidence angles ranging from 15° to 45° (ESA, 2004). These data became accessible starting from 2005. Data availability in the Arctic is high (Bartsch et al., 2009) due to overlapping swaths related to the polar orbit

and low demand of higher resolution acquisitions in these regions. All used ASAR GM level 1b data have been acquired with HH-Polarization (horizontally - with respect to the earth surface - transmitted and received). They are gridded to 500 x 500 m.

For the study area Kytalyk (NE Siberia) also ENVISAT ASAR data acquired in Wide Swath (WS) mode and HH polarization have been tested. They cover the same incidence angle range and swath but spatial resolution is finer, although still medium,

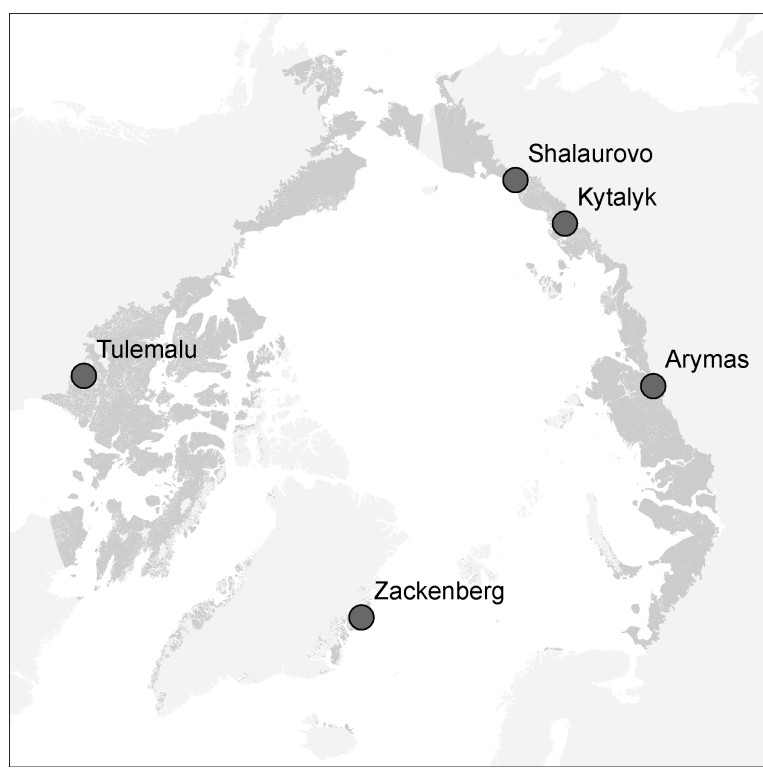

**Figure 1.** Location of field sites with high resolution land cover based soil organic carbon (Tab. 1) maps and area covered by the Circumartcic Vegetation Map (CAVM, Walker et al. (2002)) as well as ENVISAT ASAR GM data (medium grey)

with ~120 m (Closa et al., 2003). Data availability of this mode is lower since it was acquired on request only. They are commonly gridded to 75 m x 75 m (e.g. Bartsch et al., 2007; Santoro et al., 2011; Reschke et al., 2012).

## 2.2 Soil organic carbon data

In-situ measurements of SOC used in this study have been collected from five different sites across the arctic: Shalaurovo,
5  Kytalyk and Arymas in Siberia, Zackenberg on Greenland and Tulemalu in Canada (Fig. 1). All are located within the tundra biome and are characterized by continuous permafrost terrain. For these sites, the investigated SOC depth increments are 0–5cm, 0–30 cm and 0–100cm. Two types of input data are used. Soil pedon point data and maps of SOC derived from thematic up-scaling of the soil pedon data using high to very high resolution optical satellite and airborne data (Palmtag et al., 2015; Siewert et al., 2015; Hugelius et al., 2010). In order to obtain these maps collected soil pedons were grouped according to
10  the thematic classes in these schemes. Simple arithmetic means and standard deviations were then calculated for each SOC storage depth increment per thematic class (and for the calculation of 0-30 cm and 0-100 cm SOC stocks). These means were subsequently weighed by the proportional representation of each thematic class in the study area to arrive to a weighed

**Table 1.** Available SOC data upscaled from high resolution remotely sensed data

| Site name | Upscaling source | harmonized land cover classes | Source | pedon data |
|---|---|---|---|---|
| Kytalyk | QuickBird | grass, willow, fen, other, tussock | Siewert et al. (2015) | 21 |
| Zackenberg | Airborne Hyperspectral | grass, heath, willow, fen, fell, boulder, other | Palmtag et al. (2015) | 24 |
| Shalaurovo | QuickBird | grass, willow, fen, tussock | Palmtag et al. (2015) | 18 |
| Tulemalu | Landsat 7 ETM+ | dry, moist and wet tundra, fen, bog | Hugelius et al. (2010) | 35 |
| Arymas | QuickBird | grass, willow, trees, fen, dry tundra | Palmtag et al. (in review) | 35 |

'landscape-level' mean for the entire study area. Statistical uncertainties in this type of approach have been described in Hugelius (2012).

Fourteen thematic classes (partially with subclasses for grasslands and fens) based on the classes from the local land cover classifications are distinguished across the Arctic for this study. The SOC stocks in the upper 30 and 100 cm of the soil for certain classes from different sites differ from each other since they have been adjusted site by site. SOC values range from almost 0 kg m$^{-2}$ at alpine and barren ground locations to more than 80 kg m$^{-2}$ for peat bogs. The maximum of non peat sites is approximately 35 kg m$^{-2}$ for 100 cm and 15 kg m$^{-2}$ for 30 cm. Table 1 provides further details and the data sources and land cover map thematic content.

The Northern Circumpolar Soil Carbon Database (NCSCD) by Hugelius et al. (2013, 2014) provides SOC stocks in the circumpolar permafrost region. The NCSCD is a polygon-based digital database compiled from harmonized regional soil classification maps in which data on soils have been linked to pedon data from the northern permafrost regions to calculate SOC content and mass. It includes SOC values for 0-30 cm, 0-100 cm, 0 - 200 cm and 0-300 cm. For this study, only the NCSCD area North of the arctic treeline as defined in the CAVM (Walker et al., 2002) is considered.

## 3 Methodology

### 3.1 Background

Radar backscatter is dependent on sensor parameters such as incidence angle, polarisation and wavelength as well as target parameters like surface roughness and vegetation structure as well as dielectric properties (Ulaby et al., 1982). Roughness and permittivity are the governing factors in case of bare soil (Oh et al., 1992). The dielectric constant highly depends on moisture content, leading to higher backscatter values in the microwave range under wet soil conditions (Woodhouse, 2006). Regions with soil conditions close to saturation near the surface can be therefore identified using SAR data. This has been demonstrated applicable for peatland detection at high latitudes with C-band (Bartsch et al., 2007, 2009; Reschke et al., 2012). The wet and at the same time high SOC areas have a low bulk density over several tens of cm and are water/ice rich (more than 60% at e.g. Kytalyk, Weiss et al. (2015)).

The dielectric constant is significantly lower under frozen conditions. Frozen soils cause therefore similar backscatter like dry soils which has been specifically exploited for C-band applications (e.g. Wagner et al., 1999; Park et al., 2011). Winter backscatter is thus determined by the above surface remains of vascular plants, surface roughness, near surface soil texture and, if present, also snow cover. The latter has little influence in early winter (Naeimi et al., 2012). Interaction of C-band signals with snow are lower than for shorter wavelengths (Ulaby and Stiles, 1981). The signal may also penetrate a few cm into the soil. In undisturbed environments (no buildings or agriculture) it can be assumed that scattering is governed by soil type and vegetation cover. The influence of vascular plants on signal interaction is however limited at C-Band (approximately 5.6 cm wavelength, Waring et al. (1995)). Surface roughness thus plays an important role for spatial differences in backscatter during frozen conditions in tundra regions. Specifically data acquired in HH (horizontally sent and received) polarization are expected to represent soil conditions better than VV (vertically sent and received) polarization (Brown et al., 2003). Vertically (with respect to the Earth surface) polarized waves interact more with vertically structured vegetation parts (stems) than horizontally polarized waves. HH as well as HV polarizations are thus more sensitive to roughness than VV polarizations (Holah et al., 2005).

It has been shown for C-band (Jagdhuber et al., 2014) that volume scattering (at anisotropic particles) dominates for peatland soil during unfrozen conditions and it changes to surface scattering when frozen. The dielectric contrasts between scattering components decrease and surface roughness indeed determines the magnitude of backscatter.

Tundra and in general wetland environments are commonly classified based on non-frozen period data when SAR data are employed. There are to date only very few studies which make use of frozen period acquisitions (Duguay et al., 2015; Widhalm et al., 2015). The advantage for using winter data is that only roughness and volume scattering contributes to the return signal intensity. During summer, there is the influence of liquid water in addition. High C-band backscatter areas are therefore often open wetlands (especially peatlands, e.g. Bartsch et al. (2009); Reschke et al. (2012)) but can be also areas with high roughness and/or volume scattering. Locations with higher Soil Organic carbon (SOC) are areas with low roughness (with respect to C-band, 5.6 cm wavelength). They have a smoother surface (with respect to the 5.6 cm wavelength) than drier low carbon sites in the high arctic what leads to the hypothesis that C-band backscatter can be used as proxy for SOC content (Fig. 2).

Interaction of the C-band signal with snow grains needs to be accounted for. There is especially an impact when ice crusts form (Naeimi et al., 2012; Bartsch, 2010). Backscatter does increase in such cases. C-band is however less sensitive to snow pack changes than shorter wavelengths (such as e.g. Ku-band, Bartsch (2010)). Backscatter can increase during the course of the winter by about one db at some locations (Naeimi et al., 2012). In a case study for Yakutia, no increase of ASAR GM backscatter with increasing SWE (snow water equivalent) could be observed (Park et al., 2011). In order to account for possible contributions by snow cover

1. only December data should be used, assuming that there are frozen conditions and snow depth is still limited, and

2. the minimum from as many as possible years should be calculated in order to have the lowest as possible impact (this also accounts for the GM specific noise)

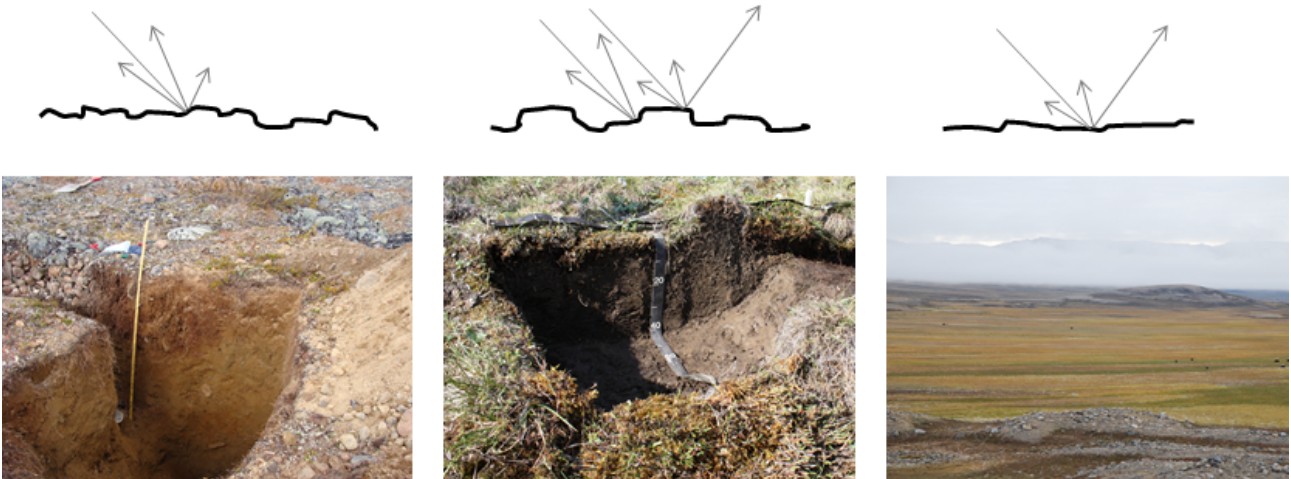

**Figure 2.** Top: Sketch of scattering from different surfaces. The length of the arrows represents the backscatter intensity: left - rough surface with e.g. pebbles at the surface, middle - surface with partial high and low signal return, right - smooth surface with almost specular reflection (adapted from ESA 2004); Bottom: Photographs (G. Hugelius) representing different roughness and soil carbon types. Left: Soil pit in carbon poor landscape, middle: hummocky carbon rich soil profile across the active layer of a frost boil, right: carbon poor slope in the front and carbon rich valley in the background (Zackenberg).

## 3.2  SAR data processing

First, automatic geocoding and radiometric calibration were performed. The SAR Geophysical Retrieval Toolbox (SGRT, Vienna University of Technology, Sabel et al. 2012) was used for the required preprocessing of the ENVISAT ASAR GM level 1b data. This is a collection of routines which manages SAR geocoding and radiometric calibration by calling other non-commercial and commercial software packages. By incorporating orbit information (DORIS (Doppler Orbitography and Radiopositioning Integrated by Satellite) orbit files) and digital elevation data (Shuttle Radar Topography Mission-improved U.S. Geological Survey GTOPO30 digital elevation model), geocoded images were produced with sub pixel accuracy (Park et al., 2011; Pathe et al., 2009). The data are resampled into a fixed 15 arc-seconds grid (datum WGS-84), within 0.5° by 0.5° tiles, to allow efficient spatial and temporal analysis. The data (> 8000 scenes north of 60° N) was normalised to a reference angle of 30° by fitting a linear model to the backscatter data (Pathe et al., 2009; Sabel et al., 2012) in order to remove the influence of local incidence angle on radar backscatter. The model provides an estimate of the slope in units of decibel per degree of incidence angle which characterizes the decrease of the radar backscatter from near range to far range. The model is calibrated for each pixel separately using the acquisitions from overlapping orbits (Wagner et al., 2008). No data can be processed with the tools used for orthorectification (SGRT) for scenes which cross the dateline with this toolbox. This leads to a data gap in far Eastern Russia. The dataset has been eventually resampled to a grid with polar stereographic projection with 500 x 500 m pixel size.

On average 45 December acquisitions have been available per pixel. Since GM data exhibit a comparably high noise (Park et al., 2011) temporal and/or spatial statistical measures (averaging, filtering etc.) need to be applied.The mean value could be used in order to account for noise alone. There are however also other effects that need to be accounted for, especially snow related changes such as the formation of ice layers due to rain on snow. They would increase the backscatter, but are not expected to be present in all years at that time of the year. The usage of the minimum backscatter value (from several years) reduces the probability that structure change affects the backscatter dataset used for SOC retrieval. The minimum of the entire record for each pixel was therefore calculated in this study instead of single values representing a certain date. Summer (Juli and August) data have been processed in addition for the Kytalyk site and mean values derived. This dataset is used to exemplify the advantage of using winter data opposed to summer records. Data are derived as sigma nought and converted to dB. The dataset has been masked for lakes and glaciers based on the map classes of the Global Lakes and Wetlands Database (Lehner and Döll, 2004) and GlobCover (Bicheron et al., 2008) as well as tree line (Walker et al., 2002).

### 3.3 Determination of relationship between backscatter and SOC

The C-band backscatter is directly compared to locally up-scaled SOC maps and underlying pedon (point) data (table 1, Fig. 1). Not all classes and the full range of SOC values can be found at single sites. A region with lower SOC (Zackenberg) and a site with medium to high values (Kytalyk) are therefore used in combination to obtain a representative range for the establishment of the empirical relationship for up-scaling. The maps of the remaining sites have been used for validation.

Zonal mean values (a zone refers to a land cover class) have been extracted for the SOC classes available for Kytalyk and Zackenberg for model calibration. The advantage of the zonal mean opposed to the pedon data is the scale comparability to the GM data.

The Pearson correlation has been derived for the zonal means and eventually a function determined by least-squares regression. The obtained function has been subsequently applied to the circumpolar dataset. The land cover based SOC maps available from Tulemalu, Arymas and Shalaurovo have been used for validation. Regional differences have been assessed using the soil map based NCSCD (v2.2) by Hugelius et al. (2013). It has been converted to a 500 x 500 m gridded dataset with separate layers for each SOC class and percentages of the soil types turbel, histel and histosols. This dataset has not been applied for training since its based on different types of data sources around the Arctic. The impact of soil type on the SOC retrieval is in addition investigated using the information available from the original pedon (point) information from all study sites since this information is not preserved in the land cover classifications.

### 3.4 Circumpolar evaluation

The validity of the approach to the tundra area is also assessed with satellite records of vegetation (NDVI) and unfrozen period length (as obtained from Metop ASCAT, Naeimi et al. (2012); Paulik et al. (2012)). MODIS NDVI data were retrieved from the online Data Pool, courtesy of the NASA Land Processes Distributed Active Archive Center (LP DAAC), USGS/Earth Resources Observation and Science (EROS) Center, Sioux Falls, South Dakota, https://lpdaac.usgs.gov/dataaccess/datapool. The NDVI records have been re-classified to represent ranges of 0.05 for each class. The number of unfrozen days has been

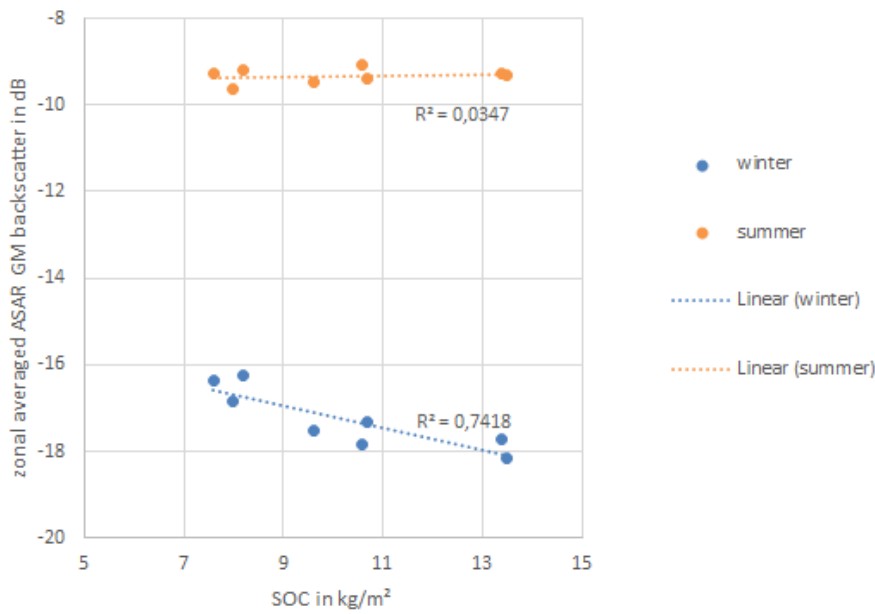

**Figure 3.** SOC from land cover classifications (0-30cm) versus winter and summer backscatter from ENVISAT ASAR GM for Kytalyk

aggregated for each available year (2007-2013) and averaged for the available time period. The average SOC values from the GM approach as well as from the NCSCD have been extracted for each NDVI and unfrozen period length class. Wetness level classes based on C-band radar backscatter ranges as defined in Widhalm et al. (2015) are also converted into SOC and discussed.

## 3.5 Transfer of the approach to WS data

The higher spatial resolution (but lower sampling rate and inconsistent coverage) data from ENVISAT ASAR Wide Swath (also HH polarization) have been used in order to test the transferability of the approach across scales for the Kytalyk study site in NE Siberia. Due to the limited data availability, normalization cannot be applied as for the GM data (approach by Wagner et al., 2008; Sabel et al., 2012). Sabel et al. (2012) and Wagner et al. (2008) exploit the availability of a representative range
of incidence angles for a certain location by using acquisitions from several overlapping orbits. A conventional method which corrects for local terrain related effects only (as available with the free NEST toolbox by the European Space Agency) has been used instead. The radiometric normalization available with NEST only accounts for terrain effects. This leads to a location specific bias with respect to the circumpolar GM based dataset. It was therefore required to adjust the WS data to the value range of the GM dataset. The incidence angle differs however by less than 0.1° across the Kytalyk and Zackenberg sites. A
single offset value per site can be therefore used to adjust the WS backscatter to GM. It has been derived from the average regional backscatter of both datasets.

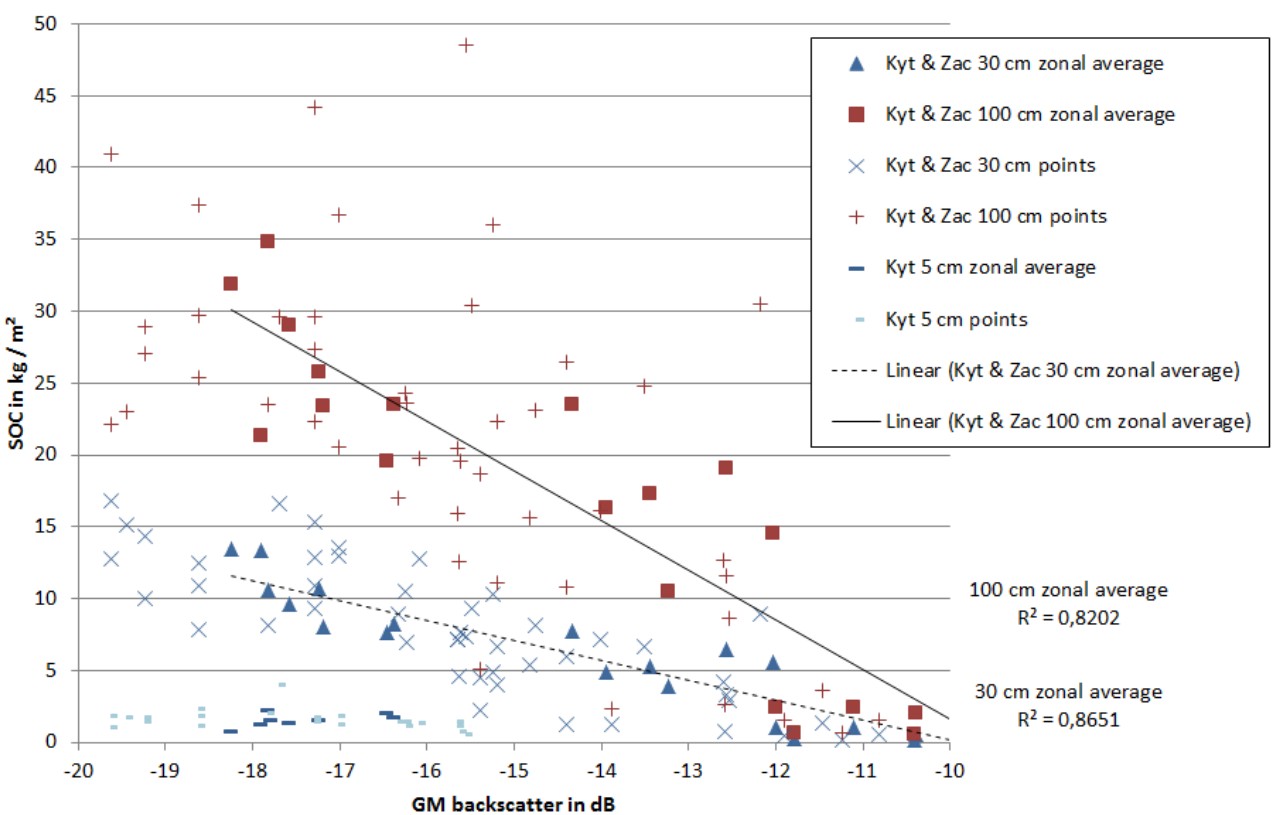

**Figure 4.** SOC from land cover classifications and pedon data (points) for all depths versus backscatter from ENVISAT ASAR GM, for Kytalik (Kyt) and Zackenberg (Zac). 5 cm data are only available for Kytalyk.

## 4 Results

### 4.1 SOC determination

No relationship ($R^2$ = 0.03) can be found for the SOC zones at Kytalyk in case of comparison to summer backscatter as soil moisture adds to the backscatter of the wetter (and at the same time higher SOC) sites. This differs for winter data. The higher

5    SOC, the lower is the winter backscatter ($R^2$ = 0.74) since it excludes the soil moisture effect (Fig. 3).

The range of dB of the GM data which corresponds to the SOC values represented in the reference datasets is about 8 dB for land cover class averages and almost 10 dB for pedon data. An R² of 0.86 was determined for the linear relationship between backscatter of GM data and SOC of 0-30 cm from land cover unit derived SOC maps of Kytalyk and Zackenberg (Fig. 4). Only 82 percent of the variation can be explained in case of SOC 0-100 cm. SOC variation for 5 cm depth can not be resolved with

10   the C-Band data.

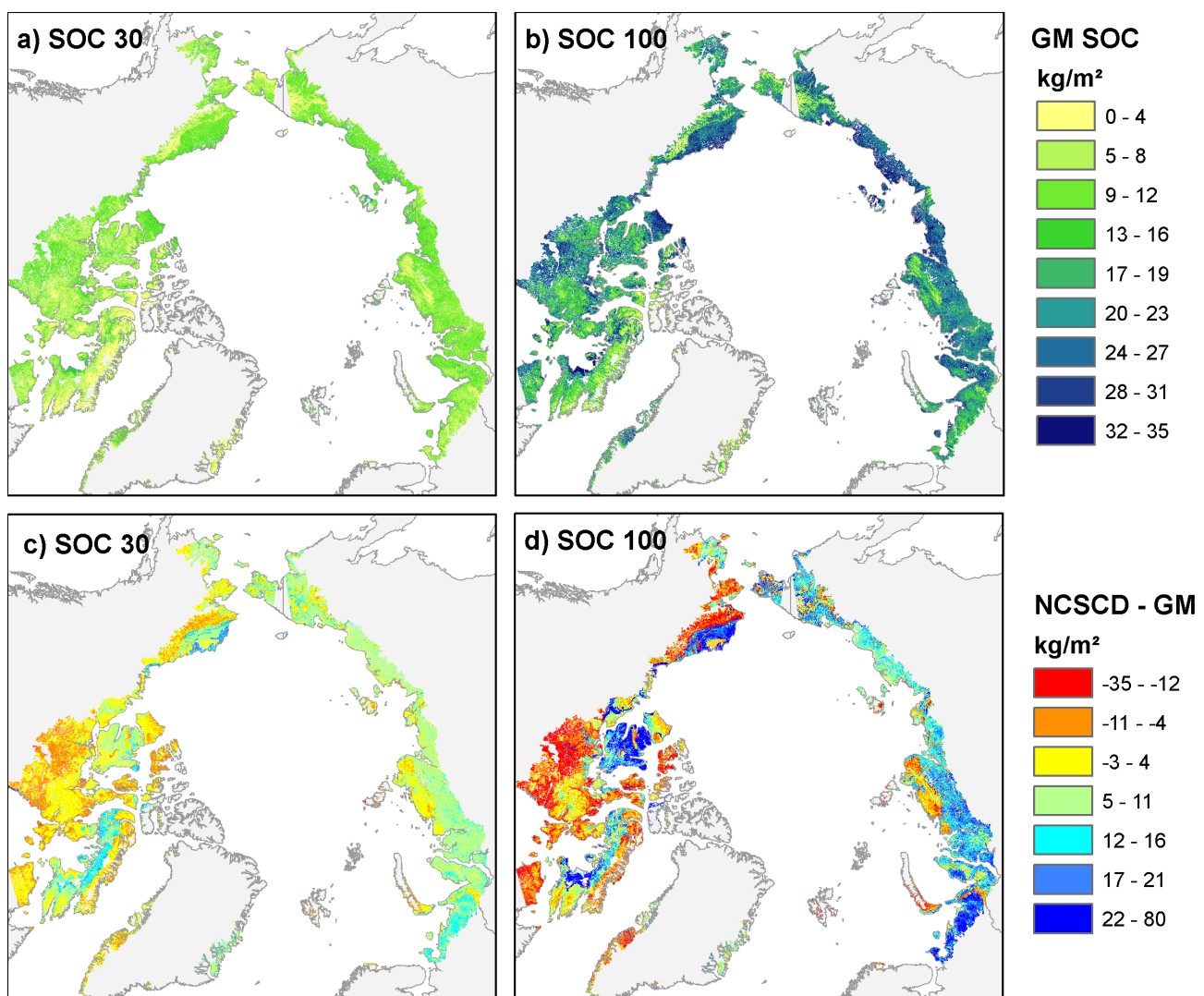

**Figure 5.** SOC results from ENVISAT ASAR GM a) 0-30 cm and b) 0-100 cm; SOC difference between NCSCD and ENVISAT ASAR GM c) 0-30 cm and d) 0-100 cm

Average and maximum SOC for the CAVM domain reach 7.4/14 kg m$^{-2}$ for 30 cm and 19.6/36 kg m$^{-2}$ for 100cm (Fig. 5 a and b) when the linear models are applied to the entire GM dataset.

Similar coefficients of determination can be obtained using WS data (Fig. 6). The linear relationship for GM and SOC is also valid for the higher spatial resolution WS data, however, only after offset correction. Further on, it cannot be excluded that snow conditions with ice layers (leading to higher backscatter) are included in the WS sample. In case of Kytalyk, the GM backscatter values are more than 2 dB lower than in WS.

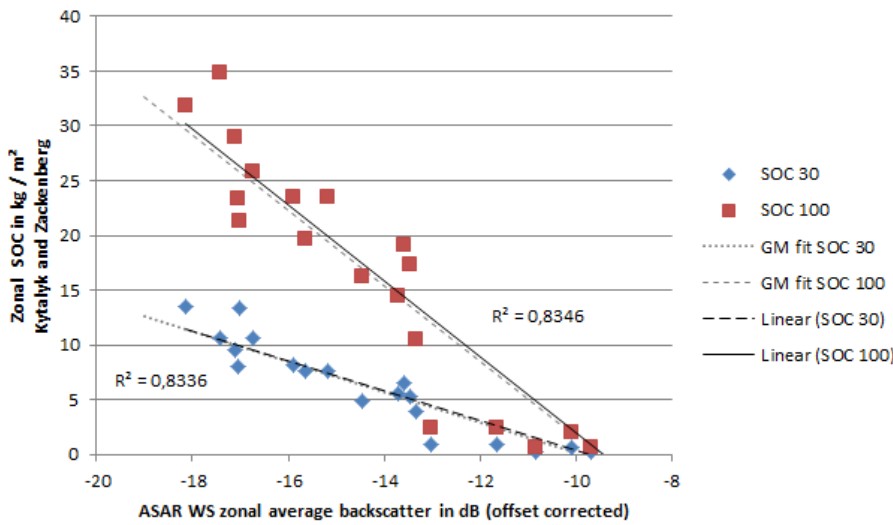

**Figure 6.** SOC for 30 and 100 cm depths versus backscatter from ENVISAT ASAR WS (offset corrected) for Kytalyk and Zackenberg including function for linear fit of the GM data (see Figure 4)

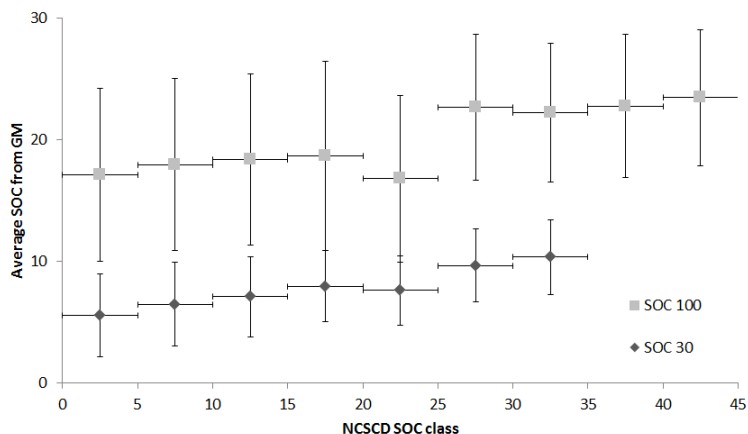

**Figure 7.** SOC in kg m$^{-2}$ from the GM results averaged over a class range of 5 kg m$^{-2}$ in the NCSCD

RMSE determined using the Tulemalu and Arymas reference (point) data is 7.67 kg m$^{-2}$ for 0–30 cm and 17.24 kg m$^{-2}$ for 0–100 cm. If peat bog sites are excluded values are reduced to 3.79 kg m$^{-2}$ and 7.58 kg m$^{-2}$ respectively. This corresponds to 20-25% of the range of in-situ SOC values. This is in the order of the standard deviation found for SOC values within the specified land cover classes at e.g. Zackenberg (Palmtag et al., 2015). The averages of the GM results (100 cm) over the validation sites reflect the differences between Shalaurovo and Arymas (Table 2). SOC is lower at the latter site what agrees with the in situ records.

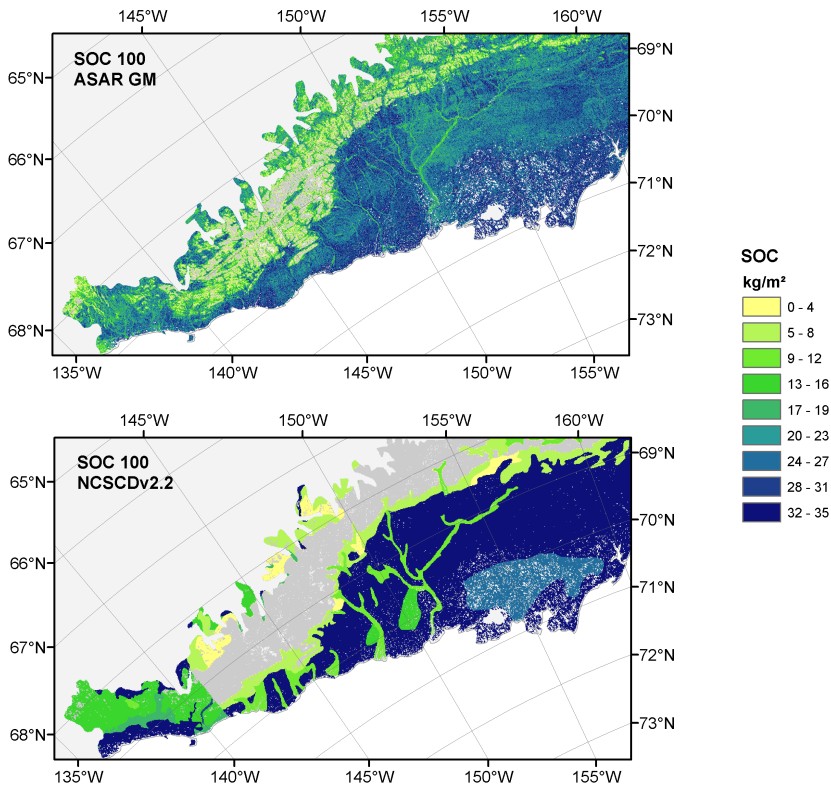

**Figure 8.** Subset maps (Alaska-Canadian border region parts of the Brooks Range and North slope) of SOC 0-100 cm results from ENVISAT ASAR GM and the NCSCD; for legend see Figure 5a, medium grey values correspond to 0 kg m$^{-2}$

**Table 2.** Averaged SOC in kg m$^{-2}$ (30 or 100 cm as indicated) for in situ (pedons - P), remotely sensed (ASAR GM - Global monitoring mode, WS - Wide Swath mode) and the NCSCD soil data. Only points with available values for pedon as well as ASAR data are used.

| Site name | P 30 | WS 30 | GM 30 | NCSCD 30 | P 100 | GM 100 | NCSCD 100 |
|---|---|---|---|---|---|---|---|
| Kytalyk | 11.2 | 10.9 | 10.5 | 17.6 | 27.4 | 27.4 | 57 |
| Zackenberg | 4.2 | 6.84 | 5.4 | 9.7 | 16.1 | 15 | 17.8 |
| Shalaurovo | 9.6 | - | 10.5 | 17.1 | 26.6 | 27.3 | 38.4 |
| Tulemalu | 13.8 | - | 9.2 | 6.3 | 28.5 | 24.1 | 19 |
| Arymas | 7.8 | - | 8.4 | 11.9 | 18.2 | 22.1 | 18.2 |

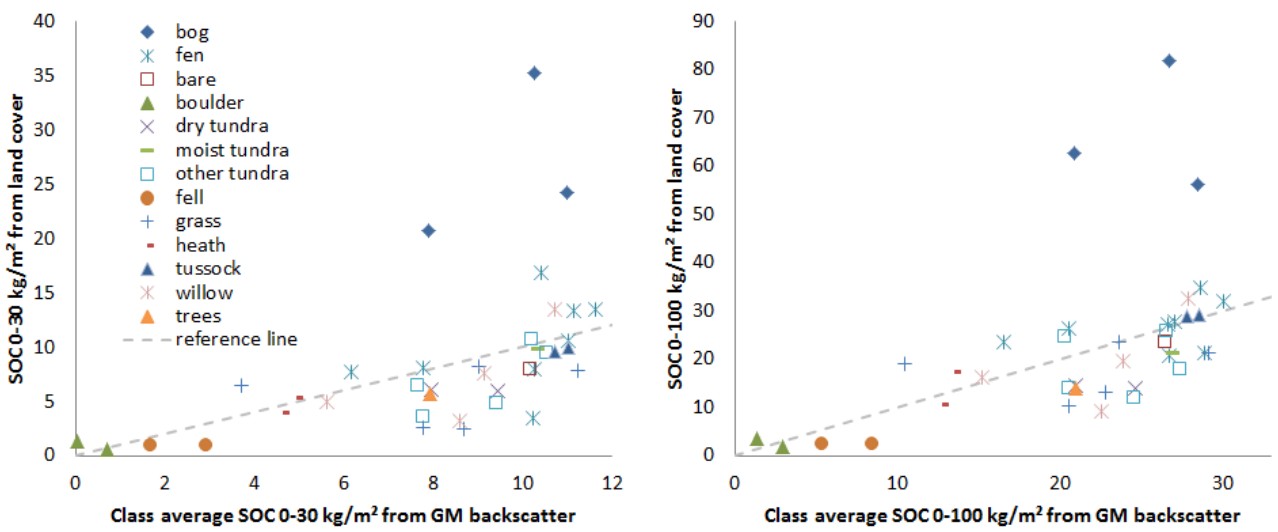

**Figure 9.** SOC from landcover classifications (all sites: Kytalik, Zackenberg, Tulemalu, Shalaurovo and Arymas) versus SOC from ENVISAT ASAR GM, by land cover (harmonized classes) and depth: left - 0-30 cm, right - 0 - 100 cm.

**Table 3.** Total SOC values (30 and 100 cm as indicated) derived from the different data sources: LC - Land cover (* excluding areas with cloud cover, approximately 5%), WS - Wide Swath, GM - Global mode and NCSCD

| Region | LC 30[*] | WS 30 | GM 30 | NCSCD 30 | LC 100[*] | GM 100 | NCSCD 100 |
|---|---|---|---|---|---|---|---|
| Kytalyk (in TgC) | 0.495 | 0.50 | 0.47 | 0.81 | 1.21 | 1.30 | 2.62 |
| CAVM domain (in PgC) | - | - | 29.2 | 41.5 | - | 80.6 | 94.0 |

### 4.2 Comparison with independent datasets

Large negative deviations of more than 10 kg m$^{-2}$ from the high resolution land cover maps (Table 1) are only found for peat bogs (Fig. 9) which are located at Tulemalu. This is also consistent with the pedon derived information for soil types (Fig. 10). SOC stocks at sites with histels are in most cases underestimated.

5    The mean difference between the NCSCD and the GM result is 3.8 kg m$^{-2}$ and 5.8 kg m$^{-2}$ for 30 and 100 cm respectively (standard deviation 6.3 kg m$^{-2}$ and 15.1 kg m$^{-2}$). SOC totals within the CAVM domain are listed in Table 3.

The differences increase with increasing SOC in the NCSCD (Fig. 7). SOC values from GM for both 30 and 100 cm are mostly higher across Northern America and lower across Siberia (Fig. 5). Transitions between areas of positive and negative value regions are sharp reflecting boundaries of maps which underlie the NCSCD (Fig. 8). SOC values change at country

10   borders, e.g. between US and Canada along 141° W. Gradients are only to a certain level of detail represented.

The residual plots (Fig. 11) for the depth of organic layer and cryoturbated carbon also confirm that the SAR method is biased low in sites with substantial cryoturbation and deep O-horizons. The SAR method is biased high for sites with limited

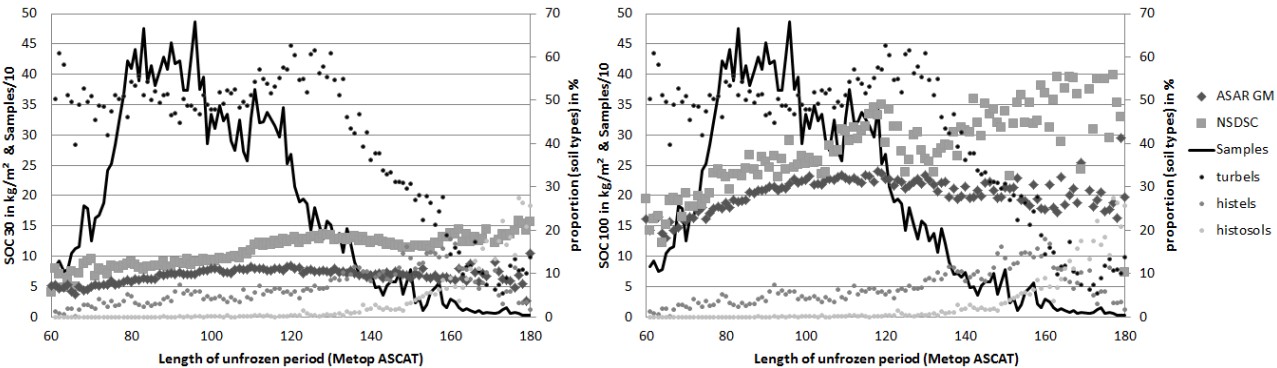

**Figure 10.** SOC from GM and the NCSCD (including mean % of turbels, histosols and histels) in comparison to unfrozen period length (source Paulik et al. (2012)). Left: 0-100 cm SOC, right: 0-30 cm. values and soil type % have been averaged for class increments of 1 day. Samples correspond to pixels of 12.5 x 12.5 km

cryoturbation and/or less than 10 cm organic layer thickness. The differences do not relate to cryoturbated carbon in case of 30 cm estimates ($R^2$=0.14), but to some extent for 100 cm values ($R^2$=0.5).

The results obtained from the NCSCD for the NDVI classes suggest a bi-modal behaviour with the first maximum for NDVI of 0.3, a local minimum for 0.45 and a second maximum for 0.7. The majority of pixels between an NDVI of 0.4 and 0.5 is located within the Canadian Arctic. GM derived SOC is higher than in the NCSCD over large parts of this region in contrast to what is observed to Siberia.

GM and NCSCD averages for the length of unfrozen period classes (Fig. 10) differ from each other. Maximum SOC in GM corresponds to about 110-120 days of unfrozen period length. A local maximum can be also found for NCSCD over that period but SOC is higher for more than 150 days.

In case of the NCSCD as well as the GM records, an increase of SOC with increasing length of the unfrozen period can be shown (Fig. 10). The variability increases for unfrozen period lengths over 120 days due to substantially low number of samples.

SOC over 0–30 cm for the Kytalyk map extent amounts to 0.5 TgC for GM (mean of 10.7 kg m$^{-2}$, standard deviation 1.7), what agrees to optical data results (0.495 TgC). The values obtained from WS data are similar with 0.47 TgC (mean 9.7kg m$^{-2}$). The standard deviation is higher with 4.4 kg m$^{-2}$. General spatial patterns in the GM and WS maps are similar to the Quickbird based results (Fig. 12, Table 3). The river flood plain shows lower SOC than the thermokarst landscape to the north. Drained lake basin patterns and associated gradients are still captured with WS but not with GM. These differences are not captured in the NCSCD database. NCSCD SOC 0–30 cm over the complete area is 17 kg m$^{-2}$ what adds up to 0.81 TgC.

The same satellite data source as in this study has been used for discrimination of wetness levels by Widhalm et al. (2015). Areas with backscatter below -16.5 dB have been shown to correspond to wet areas with potentially higher methane emissions. These areas would correspond to SOC 0-100 cm larger than 25 kg m$^{-2}$. This is confirmed by the fen and moist tundra records from the reference datasets (Table 1, Fig. 9). Medium or mixed wetness corresponds to about 20-25 kg m$^{-2}$. Several willow,

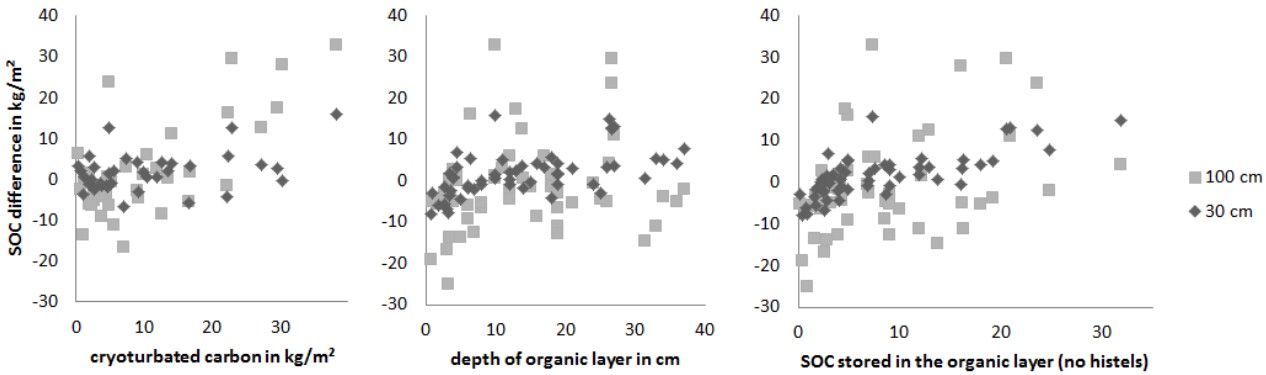

**Figure 11.** SOC differences between results from ENVISAT ASAR GM and pedon (point) measurements from Kytalyk and Tulemalu in comparison to amount of cryoturbated carbon (left; R² 0.5 and 0.14 for 100 and 30 cm respectively), depth of organic layer (middle; R² 0.59 and 0.62 for 100 and 30 cm respectively) and SOC stored in the organic layer (right; R² 0.66 and 0.69 for 100 and 30 cm respectively).

grass, fen as well as dry tundra samples fall into this category. Dry areas as defined in Widhalm et al. (2015) correspond to mostly fell, heath and boulder classes. The majority of pedon records from histels and non-permafrost mineral soils can be also found in this category (see Fig. 13). Turbels and orthels can be associated with the mixed and wet classes with about 50% of the turbels and 25% of the orthels in the wet class based on the pedon data. Orthels are also represented with about 50% in the
wet class using the GM quantification.

## 5   Discussion

### 5.1   Representativeness of C-band backscatter

The observed GM backscatter range for frozen conditions of 10 dB for terrestrial surfaces provides sufficient sensitivity to SOC variations in this landscape type. It is larger than for other common C-band backscatter based applications. In comparison,
freeze/thaw detection algorithms rely on a difference of about 1-3 dB between frozen and unfrozen conditions (Park et al., 2011). On average, backscatter decreases about 1 - 1.5 dB during freeze-up for wetlands compared to dry land cover types with about 0.5 dB over the CAVM domain (Widhalm et al., 2015). A differentiation of wetland types (Widhalm et al., 2015) from December minimum backscatter is based on class ranges starting at only 1.5 dB. Potential backscatter increase due to snow property changes is also lower than the observed range for SOC with about 2-3 dB (Naeimi et al., 2012). Soil moisture
variation (dry to saturated) in tundra regions without significant proportion of water bodies causes about 5 dB variation during summer time (Högström et al., 2014).

The empirically derived function for ASAR GM is also applicable to WS when incidence angle effects are accounted for. The location specific normalization applied for the GM records to solely December data ensures that the underlying samples represent the same surface type and condition. The number of available WS records is, however, in general lower than for GM

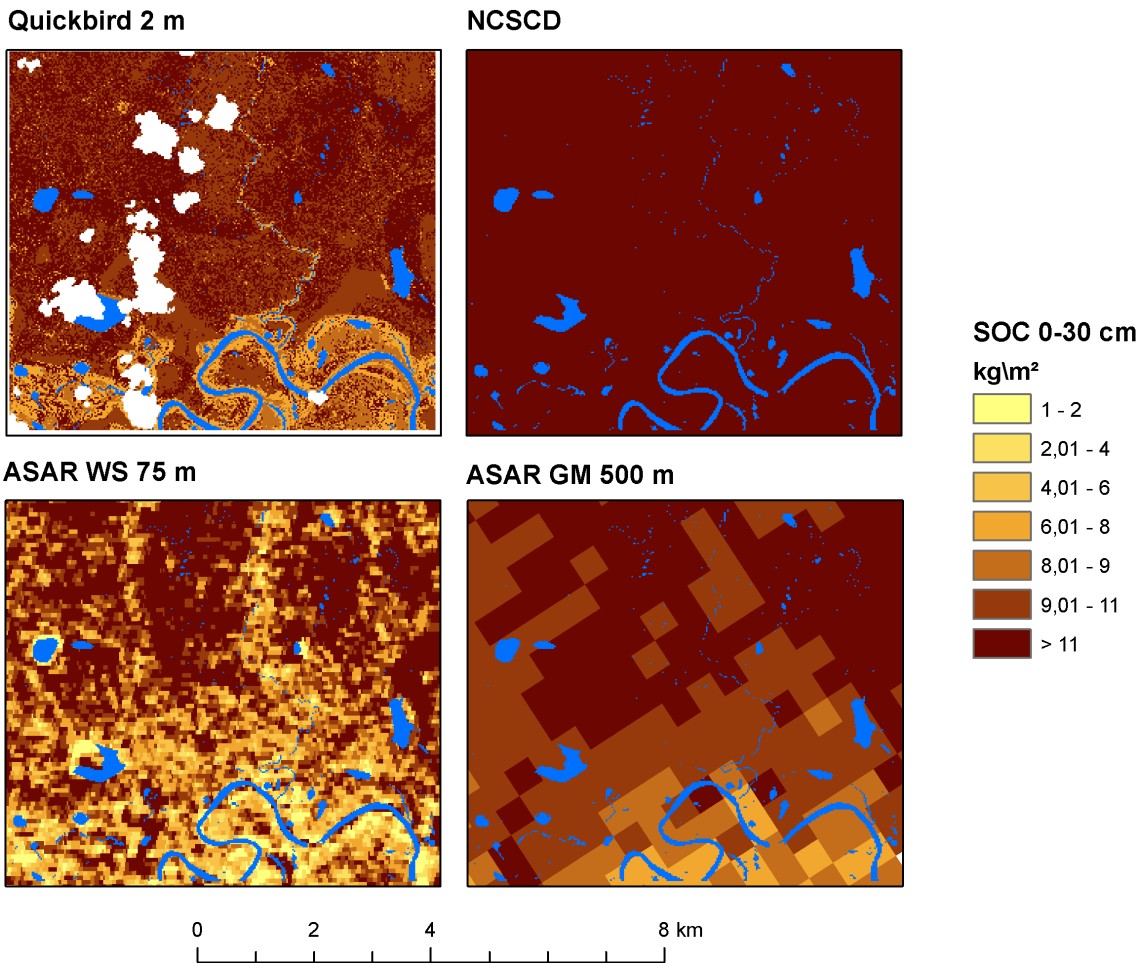

**Figure 12.** SOC 0-30 cm from Quickbird (Siewert et al. (2015)), NCSCD (Hugelius et al. (2014)), ENVISAT ASAR WS and GM (resampled to a grid with polar stereographic projection) for Kytalyk. Water bodies and clouds from Quickbird classification in blue and white respectively. NCSCD contains a constant value of 17 kg m$^{-2}$ across the area.

in the Arctic and varies spatially and temporally (Bartsch et al., 2012). When just a few images are available, as in most cases for WS, only conventional normalization can be applied. The local multi-annual backscatter minimum of December (derived to reduce impact from a potential unusually large snow depth or ice layers) might be therefore different between WS and GM. GM data are in addition characterized by much larger noise than data acquired in WS mode (Park et al., 2011). This effect is however reduced by using the minimum of the seven year record (Widhalm et al., 2015). The noise may still contribute to the slightly larger range of values observed in the GM data (Fig. 4 and 6).

The for SAR acquisitions typical foreshortening effects would need to be accounted for when this method is applied in mountain areas, especially for WS data. An application of the approach to WS over larger areas would also require appropriate normalization in order to account for incidence angle effects. The utilization of the effect of incidence angle on sensitivity to

roughness effects (Baghdadi et al., 2001) could be exploited in addition. The currently operating Sentinel-1 satellite which is a follow on mission of ENVISAT ASAR also acquires data in C-Band. The polarizations vary (VV and VH is common). For this study, only HH has been available. The sensitivity to SOC may differ for other polarizations and thus impact the transferability of this approach to e.g. Sentinel-1.

redIn cases where the near surface soil is close to saturation during summer, C-band can be used to distinguish peatlands to some extent (Bartsch et al., 2009; Reschke et al., 2012). This does however only lead to a yes/no classification. Such maps (or any other appropriate landcover classification) could be used in addition to the presented approach in order to indicate areas where it is expected that SOC is underestimated.

The correlation with winter backscatter is expected to result from a combination of roughness (surface response) and volume
scattering within the remains of the vegetation (regarding snow, see below). In order to distinguish the different scattering types, polarimetric SAR data as e.g. used in Ullmann et al. (2014) would be required. Such data are however not available from ENVISAT ASAR GM. Since winter data are used, only interaction with the remaining woody parts is expected. The contribution from volume scattering from woody vegetation becomes important when stems reach a certain size with respect to the used wavelength. The used training and validation sites include also willow dominated landcover. The obtained results
from these locations do not indicate that the chosen approach is not applicable. SOC derived from ASAR GM is close to SOC from high resolution optical data (Figure 7) for willow classes. SOC might be however underestimated in case of thicker stems (more than 5 cm).

L-Band (approximately 23cm wavelength) is expected to better penetrate to the ground in the tundra-taiga transition zone. It may in general give better indications of soil moisture during the summer season. The interaction with the surface material
(roughness and volume scattering) is however expected to be much lower than with C-band with respect to the tundra surface characteristics (see Fig. 2). The sensitivity to the relevant surface features which are used as proxy for SOC is expected to be lower at L-band.

## 5.2   Evaluation results

The large difference (about half of the values in the NCSCD) of SOC observed between the up-scaled maps (both from optical
as well as the radar data, Figure 12) at Kytalyk is similar to the NCSCD-GM differences throughout the entire Arctic (Fig. 7). Such a deviation is, however, not found when GM is compared to pedon data across all sites (Fig. 9). NCSCD values are also considerably higher than in situ records for the Siberian sites (Table 2).

The difference between deviations of SOC estimates from the NCSCD for both 30 and 100 cm between the Siberian and North American part can be to some extent explained by the presence of peatlands. The SAR approach underestimates SOC for
histels as well as histosols (Fig. 14). Similarly SOC from the GM datasets is lower when the proportion of turbels exceeds 20%. This applies to the 30 cm as well as 100 cm results. Large parts of the Canadian Arctic as well as the Brooks Range in Alaska have, however, higher SOC in the GM results than in the NCSCD. This might be due to inconsistencies in the underlying maps. E.g. most of the Alaskan Brooks Range has a value of zero SOC (Fig. 8).

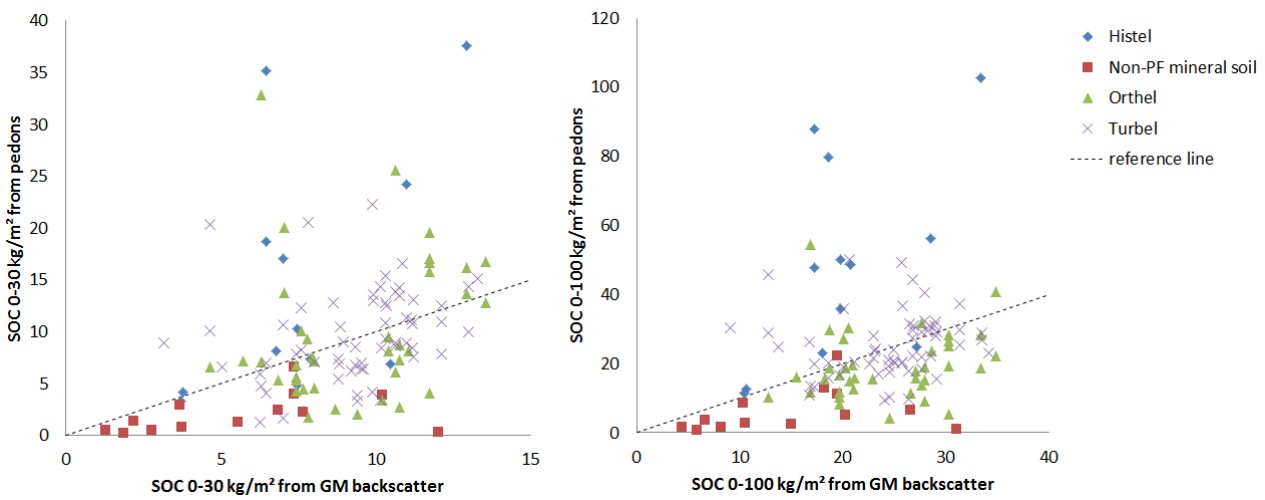

**Figure 13.** SOC 0-30 cm (left) and 0-100 cm (right) results from ENVISAT ASAR GM in comparison to pedon (point) measurements from Kytalyk, Zackenberg, Shalaurovo, Arymas and Tulemalu by soil type.

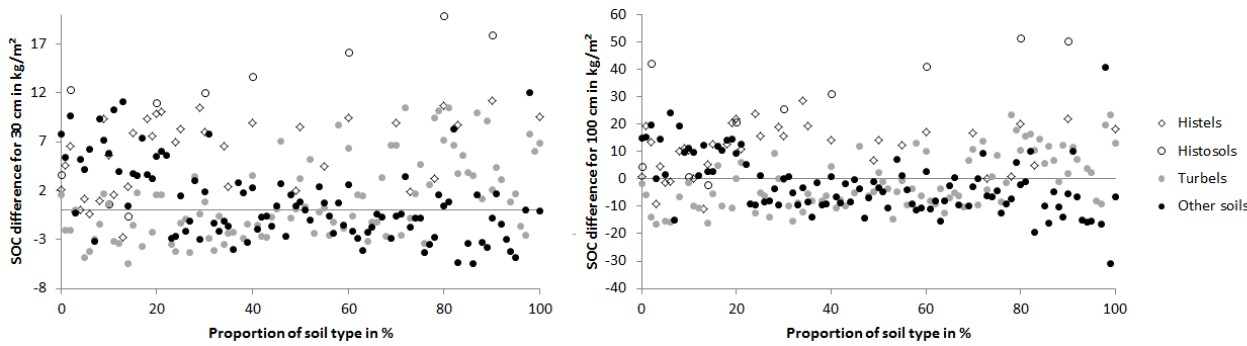

**Figure 14.** Averaged differences of SOC in kg m$^{-2}$ between NCSCD and GM by soil type (averaging intervals 1%, source NCSCD).

Deviations from the land cover based SOC data (Fig. 9) could be also partially due to the limited availability of the reference dataset (Table 1). The included data represent only a small area within the high resolution satellite data classifications of several kilometer extent. The available evaluation datasets (Tulemalu and Arymas) do not provide the full range of possible SOC values over areas which are large enough to be detected with the spatial resolution of the SAR data. Validation data only represent

5    SOC values above 10 kg m$^{-2}$ over 100cm. Only medium to high SOC values are thus used for the determination of the RMSE. A larger range may lead to a reduction of the RMSE.

The larger standard deviation in the WS derived SOC stocks compared to GM for Kytalyk is expected due to its higher spatial resolution. This reflects the complexity of the landscape at Kytalyk (Siewert et al., 2015). Future studies should consider measures of variation within the SAR resolution cell when using coarse data such as ASAR GM or make use of higher

10   resolution SAR as demonstrated applicable for WS.

The high coefficient of determination for 30 cm as well as 100 cm implies that a relationship exists between SOC accounted over these horizons. This agrees with the NCSCD. Based on linear correlation of the NCSCD version 2.2 pedon database (Hugelius et al. (2013), n=523, p<0.001, log transformation of data) the R between 30 cm and 100 cm SOC stocks is 0.78. The applied linear relationship between backscatter and SOC is not valid for high carbon areas such as peat soils. Surface properties as seen by C-band SAR are not changing for SOC 0-100cm values higher than approximately 35 kg m$^{-2}$. Peat deposits can have similar surface properties, but variable depths. Peat accumulation is related to age and the accumulation rates decrease exponentially in thermokarst basins (Jones et al., 2012).

SOC values are also underestimated in areas with extensive cryoturbation, as e.g. in North America (like Northern Alaska and the coast of the Canadian Archipelago, Fig. 5 c and d). Soil types are not reflected in the reference maps (Table 1) but in the original pedon data. A comparison with the in situ records shows that SOC is underestimated in most cases for histels (Fig. 13), which are equivalent to permafrost peatlands (>40 cm O-horizon). Also turbels (in this study defined as permafrost soils with more than 1 kg cryoturbated carbon) are underestimated in some cases. This can be clearly observed for SOC over 100 cm. The process of cryoturbation may lead to a roughening of the surface and thus ambiguities with characteristics typical for lower SOC content soil types.

SOC of non-permafrost mineral soils and orthels (mineral permafrost soil without cryoturbation) is mostly overestimated although the assumption that carbon content increases with decreasing backscatter seems to be valid. This could be due to the inclusion of areas with cryoturbated soils into the training dataset. Turbels are typical at Kytalyk. Further on, the occurrence of mixed pixels may contribute to this overestimation. Non-permafrost mineral soils are often found at river banks and their covered area is smaller than the resolution of ASAR GM data. Backscatter within such a GM pixel would be lower in case of grounded ice within that cell. This is partially possible for shallow river sections. There is interaction of microwaves at the water – ice boundary of floating ice. If ice on water bodies is freezing to the ground, the backscatter mechanisms change and the backscatter intensity recorded at the sensor drops significantly (e.g. Jeffries et al., 1993). This has an effect on SOC retrieval in case of pixels which include very shallow water which is freezing to the bed already in early winter. It results in higher SOC estimates. A value of more than 30 kg m$^{-2}$ over 100 cm is e.g. derived from GM for a pedon with 1 kg m$^{-2}$ from Arymas which represents a sand bar. A water surface map with higher spatial resolution than the used SAR data would be required in order to mask out affected pixels.

The land cover types of the reference maps (Table 1) can be associated with a certain SOC range (Fig. 13) what supports the chosen up-scaling approaches of Siewert et al. (2015) and Palmtag et al. (2015). Distinct and with a low range of values are however only boulder areas, fells, heath, dry tundra and tussock sites. Fen, willow and grass classes require further distinction into subclasses to represent their SOC value range. This needs to be considered when landscape based up-scaling is pursued from land cover maps.

The average amount of carbon across the CAVM domain of 19.6 kg m$^{-2}$ over 100 cm is lower than the estimate of 34 kg m$^{-2}$ by Ping et al. (2008) over the North American part. This could be attributed to the presence of peatlands and deeply cryoturbated soils in this region. The carbon stock total by Ping et al. (2008) is however 98.2 PgC for Northern America alone which is higher than the circumpolar account from the NCSCD (94 PgC). Mishra and Riley (2012) obtained an RMSE of

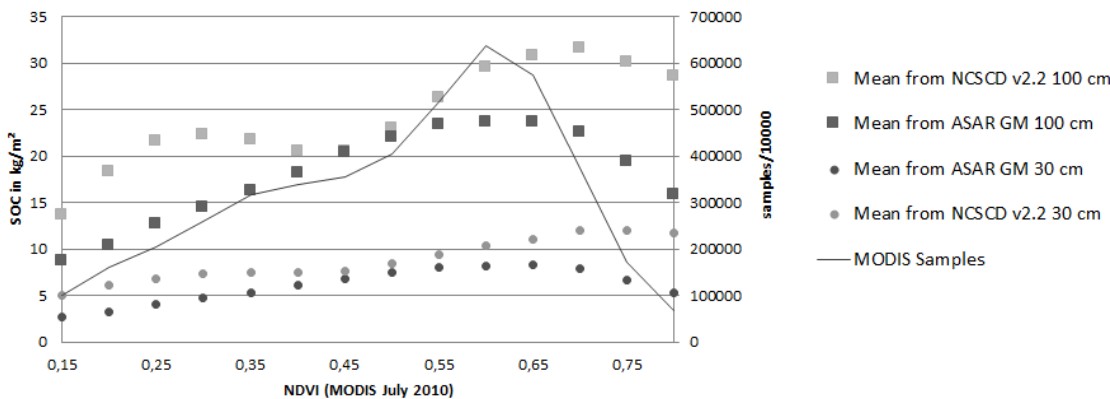

**Figure 15.** SOC from the GM results as well as the NCSCD in comparison to Normalized Difference Vegetation Index (NDVI, source: MODIS). SOC values have been averaged for class increments of 0.05 NDVI

17.8 kg m$^{-2}$ for the active layer in Alaska using environmental parameters including topography and temperature. Active layer ranges from 14 to 93 cm according to Mishra and Riley (2014). This RMSE is in the same order of magnitude as for the C-band approach (RMSE of 17.24 kg m$^{-2}$ over 100 cm).

### 5.3 Potential and limitations of the C-band approach

The conclusion of Horwath Burnham and Sletten (2010) that NDVI might be only applicable for up-scaling in the high Arctic is confirmed when GM SOC values are compared to NDVI (MODIS July 2010, Fig. 15). SOC increases with NDVI, but decreases when a certain level is reached. This differs between the GM and NCSCD values. In case of GM, values increase linearly until NDVI values of 0.6. They decrease for higher NDVI values. As SOC increases with decreasing backscatter, the same can be assumed for low to medium NDVI. A linear relationship between C-Band backscatter at HH polarization and

NDVI has been also found by Wang et al. (2013) for values higher than 0.6 for subtropical regions. They report increasing NDVI with increasing backscatter what agrees with our findings. As described above, C-band backscatter relates to higher woody vegetation in areas south of the treeline. Winter backscatter spatial variability in forested regions represents growing stock volume (Santoro et al., 2011). This is reflected in the NDVI as well as unfrozen period comparisons.

GM derived SOC indicates a maximum of SOC production in areas where there is reasonable plant productivity and litter

input and continuous permafrost promoting soil cryoturbation. This corresponds to an unfrozen period length less than approximately 110-120 days. NCSCD as well as GM SOC values increase with in increasing unfrozen period length below that length. Both calibration sites (Kytalyk and Zackenberg) fall into that zone of similar behavior. This suggests an applicability of the GM approach to areas with up to 4 months unfrozen conditions. An increase of SOC with mean annual air temperature until a certain threshold followed by an inverted relationship similar as found for the comparison of ASAR GM derived SOC

and the length of unfrozen period has been reported by Mishra and Riley (2012) for Alaska.

The local SOC minimum in the NCSCD around a length of unfrozen period of 130 days corresponds to Western Alaska and Western Russia. The peak and high differences in SOC between the datasets around 120 days does also correspond to the maximum in turbel occurrence. This confirms the limitation of the approach in areas with intensive cryoturbation as obtained from comparison with the in situ records (Fig. 13).

The wetland (wetness level) classification by Widhalm et al. (2015) which is based on the same principle (December minimum C-band backscatter) can be also interpreted for carbon levels. High SOC areas coincide with high wetness (Weiss et al., 2015). Conventional approaches for near surface saturation determination based on C-band data utilize unfrozen period data only (e.g. Wagner et al., 1999). This does however require a location specific calibration and at least one dense record of one summer season to identify high SOC areas (Reschke et al., 2012). A consistent coverage is however not available for SAR data. The sensitivity to saturation levels is also varying by vegetation coverage (Pathe et al., 2009). Such approaches further on rely on the assumption that roughness and scatting mechanisms do not change over time, what is not the case over many areas in the high latitudes (Högström et al., 2014). C-band winter backscatter can be shown to be used as an alternative, as a proxy for wetness levels as well as soil organic carbon storage.

On panarctic scale the method and dataset created currently provides in the first place a means to assess the consistency of maps from conventional sources (soil maps). In order to produce a reliable panarctic-map which fully accounts for peat, a combination with other sources (e.g. NCSCD) is required. To use the current version, areas with high SOC content (e.g. >35 kg m$^{-2}$ for 100cm) should be masked (replaced with other estimates if available). For future studies, we propose in addition a fusion with other land cover information (especially peatland extent from remotely sensed data to spatially confine the high SOC area better). This requires a reliable circumpolar peatland map, which is to date not yet available.

## 6 Conclusions

Upper limits of the applicability of the C-band SAR approach are approximately 0.6 NDVI and about 120 days of unfrozen surface conditions. Near surface soil organic carbon can be quantified with C-band SAR data for arctic and subarctic environments for non-peatlands and soils with limited cryoturbation. Results suggest that in total > 29 PgC soil organic carbon is stored in the upper 30 cm north of the tree line (CAVM domain). The ENVISAT ASAR GM circumpolar estimates for SOC are about 25% lower than the NCSCD account (including peatlands). This underestimation differs between regions and points to inconsistencies in the NCSCD. The spatial continuity of our approach allows the quantification of sparsely vegetated areas that are mapped as 0 kg m$^{-2}$ in the NCSCD, as exemplified for the Brooks Range in Alaska. The estimates of total SOC stored in upper soil layer is similar for the different investigated sources (land cover based estimate and C-band backscatter from satellite data) and across scales (different C-band resolutions) for the Kytalyk test site. The results from ASAR GM as well as in situ records suggest that NCSCD estimates for the Siberian tundra area are too high.

Carbon rich soils (> 35 kg m$^{-2}$ over 100 cm) cannot be captured with this approach. Soil processes such as cryoturbation may in addition lead to increased surface roughness and therefore underestimation of SOC when using the SAR approach. A fusion of traditional land cover information and backscatter (frozen ground and low snow cover as used in this study) may

provide a means to produce spatially consistent circumpolar estimates including peat soils. The length of unfrozen period in addition to the Normalized Difference Vegetation Index are circum Arctic available products that might be suitable supporting variables for modelling the spatial distribution of soil organic carbon.

*Author contributions.* Annett Bartsch has developed the initial concept for the study, performed all analyses on the preprocessed datasets and drafted the manuscript. Barbara Widhalm performed all preprocessing of the satellite data. Peter Kuhry, Gustaf Hugelius, Juri Palmtag and Matthias Siewert have collected the in situ data and prepared them for this analysis. All co-authors contributed to concept development and writing of respective parts of the manuscript including discussion.

*Acknowledgements.* Financial support by the European Commission (FP7-ENV-2011, Grant Agreement no. 282700) through the project Changing Permafrost in the Arctic and its Global Effects in the 21st Century (PAGE21) is gratefully acknowledged. In-situ data sharing was made possible through the same project. The Kytalyk soil organic carbon inventory was conducted with support of the above-mentioned EU PAGE21 project; Arymas and Shalaurovo soil sampling and analyses were possible through the financial support provided by the VR ESF CryoCarb project; the Tulemalu fieldwork and subsequent SOC analysis were supported through the EU GLIMPSE project (contract EVK2-2001-00337) and the Swedish Research Council (VR); the Zackenberg soil organic carbon inventory was made possible through support of the Norden Permanor and Nordforsk DEFROST projects.

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

(0)(