# Peer review of "Can C-Band SAR be used to estimate soil organic carbon storage in tundra?"

_Biogeosciences, 2016_

## Referee Comment (RC1) · Anonymous Referee #1 · 31 May 2016

Authors evaluated the use of microwave backscattering Radar data to predict the SOC stocks of Tundra regions. Authors report that the C-band of SAR data can be used to predict the SOC stocks. Validation results show that their approach under predicts the SOC stocks as it does not capture the dynamics of peatlands and cryoturbated soils. The prediction accuracy was reported to decrease with depth. The results of this study are suitable for publication in Biogeosciences. However, at present form, the manuscript uses many undefined acronyms and the language structure is not reader friendly sometimes. The manuscript will benefit from the proper English editing.

Arctic soils are covered by thick mass of scrub vegetation (herbaceous vegetation less than 5 m tall) or thick O horizon, therefore its unlikely that the microwave spectra can reach to the mineral soil surface. As the Radar microwaves do not penetrate deeper into the soil profile (P6L3-4), the predictions might be a result of pure correlations. It will

be helpful to see more descriptions of the process/mechanisms by which the backscattered spectral data is related to the soil properties such as SOC stocks. How the surface vegetation of Tundra region impacts (helps or complicates) SOC predictions?

This study used backscatter data of December month. What about the surface snow accumulation impacts on backscattering? I think in December, the spectra won't even reach to the soil surface. So basically it can't distinguish between dry and wet areas, which is the basis to estimate SOC stock (as authors describe) in this approach. What about using the spectra of summer months where may be you can find dry and wet areas at surface?

Authors used a variety of data averaging approaches for the evaluation purpose. While generating prediction accuracy or validation errors of maps, I prefer comparison of modeled results with pedon data as done in Figure 12. Please provide R2 values in figure 12.

Authors calculated RMSE values to demonstrate the prediction accuracy of generated maps (P9L10-12). Please compare the RMSE values with the standard deviation of SOC pedon data at both depths (see Mishra and Riley, 2012). This will provide the predictive ability of the adopted approach.

Mishra U., and W.J. Riley. 2012. Alaskan soil carbon stocks: spatial variability and dependence on environmental factors. Biogeosciences, 9:3637-3645, doi:10.5194/bg-9-3637-2012.

What about using L band? Many studies have shown good correlations of L band with vegetation biomass. If the C band better discerns wet and dry surface, than peatland SOC stocks should be predicted better? See P5L18-21.

Methods: Move the "background" section to Introduction

---

## Author Comment (AC1) · 9 Jun 2016

We would like to thank anonymous Referee #1 for the comments. The questions point to a number of issues which need to be better elaborated in the description of the approach in order to make it more understandable to the reader.

Tundra and in general wetland environments are commonly classified based on non-frozen period data when SAR data are employed. There are to date only very few studies which make use of frozen period acquisitions (Duguay et al. 2015, Widhalm et al. 2015). The advantage for using winter data is that only roughness and volume scattering contributes to the return signal intensity. During summer, there is the influence of liquid water in addition. Increasing soil moisture increases backscatter. High C-band backscatter areas are often open wetlands (especially peatlands, e.g. Bartsch

et al. 2009, Reschke et al. 2012) but can be also areas with high roughness and/or volume scattering. Locations with higher Soil Organic carbon (SOC) are areas with low roughness (with respect to C-band, 5.6 cm wavelength). The higher SOC, the lower is the winter backscatter (since it excludes the soil moisture effect), see also Figure 1 which exemplifies this for Kytalyk. When comparing summer average backscatter with SOC data, no relationship can be found for the SOC zones as soil moisture adds to the backscatter of the wetter (and at the same time higher SOC) sites.

In cases where the near surface soil is close to saturation, C-band can be used to distinguish peatlands to some extent (Reschke et al. 2012). This does however only lead to a yes/no classification. Such maps (or any other appropriate landcover classification) could be used in addition to the presented approach in order to indicate areas where it is expected that SOC is underestimated.

The correlation with winter backscatter is expected to result from a combination of roughness (surface response) and volume scattering within the remains of the vegetation (regarding snow, see below). In order to distinguish the different scattering types, polarimetric SAR data as e.g. used in Ullmann et al. (2014) would be required. Such data are however not available from ENVISAT ASAR GM. Since winter data are used, only interaction with the remaining woody parts is expected. The contribution becomes important when stems reach a certain size with respect to the used wavelength. Figure 2 shows a photograph of a willow tree with stems larger than the wavelength. The used training and validation sites include also willow dominated landcover. The obtained results from these locations do not indicate that the chosen approach is not applicable. SOC derived from ASAR GM is close to SOC from high resolution optical data (Figure 8 of the manuscript) for willow classes. SOC might be however underestimated in case of thicker stems.

L-Band ($\sim$23cm wavelength) would better penetrate to the ground in these environments (a relationship with biomass is expected in forests with thicker tree trunks). It may give better indications of soil moisture during the summer season. The interaction

with the surface material (roughness and volume scattering) is however expected to be much less than with C-band with respect to the surface characteristics (see Figure 2 of the manuscript). The sensitivity to the relevant surface features which are used as proxy for SOC is expected to be lower.

Interaction of the C-band signal with snow grains needs to be accounted for. There is especially an effect when ice crusts form (Naeimi et al. 2012, Bartsch 2010). Backscatter does increase in such cases. C-band is however less sensitive to snow pack changes than shorter wavelengths (such as e.g. Ku-band, Bartsch 2012). Backscatter can increase during the course of the winter by about one db at some locations (Naeimi et al. 2012). In a case study for Yakutia, no increase of ASAR GM backscatter with increasing SWE (snow water equivalent) could be observed (Park et al. 2011, Figure 8). In order to account for possible contributions by snow cover

(1) only December data are used, assuming that there are frozen conditions and snow depth is still limited, and

(2) the minimum from the 5 year period (on average 45 acquiaitions per pixel available) is calculated in order to have the lowest as possible impact (this also accounts for the GM specific noise)

Pedon data represent point locations. Due to the very heterogeneous environment it cannot be expected that they are representative for 1km areas (GM resolution). They are therefore only of limited applicability for validation. The zonal maps which have been made based on high resolution optical satellite data are used for the calibration and validation instead. These maps have been quality checked at all the sites. We have nevertheless decided to show a comparison to the pedon data, since the soil type information (which plays a role in the applicability of the approach, as shown in the comparison with the NCSCD data) is not preserved in the zonal maps. The line in Figure 12 therefore represents only the reference and is not a fitted function.

Many thanks for pointing out the publication by Mishra & Riley.

Bartsch, A., Wagner, W., Scipal, K., Pathe, C., Sabel, D., and Wolski, P.: Global Monitoring of Wetlands - the Value of ENVISAT ASAR. Global Mode, Journal of Environmental Management, 90, 2226–2233, 2009.

Bartsch, A. (2010): Monitoring of terrestrial hydrology at high latitudes with scatterometer data. In: Imperatore, P., Riccio, D. (Ed.) Geoscience and Remote Sensing New Achievements. Chapter 14, 247-262. INTECH.

Duguay, Y., Bernier, M., Lvesque, E., and Tremblay, B.: Potential of C and X Band SAR for Shrub Growth Monitoring in Sub-Arctic. Environments, Remote Sensing, 7, 9410–9430, doi:10.3390/rs70709410, 2015.

Naeimi, V., Paulik, C., Bartsch, A., Wagner, W., Kidd, R., Boike, J., and Elger, K.: ASCAT Surface State Flag (SSF): Extracting Information on Surface Freeze/Thaw Conditions from Backscatter Data Using an Empirical Threshold-Analysis Algorithm, IEEE Transactions on Geoscience and Remote Sensing, 50, 2566 – 2582, doi:DOI: 10.1109/TGRS.2011.2177667, 2012.

Park, S.-E., Bartsch, A., Sabel, D., Wagner, W., Naeimi, V., and Yamaguchi, Y.: Monitoring Freeze/Thaw Cycles Using ENVISAT ASAR. Global Mode, Remote Sensing of Environment, 115, 3457–3467, 2011.

Reschke, J., Bartsch, A., Schlaffer, S., and Schepaschenko, D.: Capability of C-Band SAR for Operational Wetland Monitoring at High Latitudes, Remote Sensing, 4, 2923–2943, 2012.

Ullmann, T.; Schmitt, A.; Roth, A.; Duffe, J.; Dech, S.; Hubberten, H.-W.; Baumhauer, R. Land Cover Characterization and Classification of Arctic Tundra Environments by Means of Polarized Synthetic Aperture X- and C-Band Radar (PolSAR) and Landsat 8 Multispectral Imagery — Richards Island, Canada. Remote Sens. 2014, 6, 8565-8593.

Widhalm, B., Bartsch, A., and Heim, B.: A novel approach for the characterization of

tundra wetland regions with C-band SAR satellite data, International Journal of Remote Sensing, 36, 5537–5556, 2015.

Widhalm, Barbara; Bartsch, Annett; Siewert, Matthias; Hugelius, Gustaf; Elberling, Bo; Leibman, Marina; Dvornikov, Yury: Site Scale Wetness Classification of Tundra Regions with C-Band SAR Satellite Data, Proceedings of the ESA Living Planet Symposium, Prague, 9-13 May 2016.
* * *
**Fig. 1.** Comparison of zonal averaged December and Summer average backscatter from ASAR GM with Soil Organic carbon values (0-30cm) for Kytalyk

[Figure]

**Fig. 2.** Salix vegetation with lens cap (diameter 5.6 cm) as scale (source: Widhalm et al. 2016).

---

## Referee Comment (RC2) · Anonymous Referee #2 · 26 Jul 2016

General comments

The authors present a study that demonstrates the potential use of C-band SAR to determine SOC content in Northern treeless arctic regions. The inferred surface roughness from the SAR data provides an observable metric that may be correlated to measured SOC values from pedon and upscaling land cover based studies. In general the method provides reliable results and at a higher spatial resolution thant the existing NCSCD dataset. The work is innovative and appears to have great promise at better resolving SOC content across permafrost areas.

Weaknesses: Where does it go from here? Do the authors believe that it can produce a reliable panarctic map or given the limitations with carbon rich soils do they believe the method should be reserved for areas with traditional landcover-based assessments? If

so, will they produce this map? Will it be available?

Specific comments:

Abstract – last sentence: This sentence is not particularly clear. In general a brief explanation/discussion on why unfrozen period and SOC should be closely related would be very helpful. This relationship is presented in results but early in the paper it would helpful to have more detailed context on the why this is important.

Figure 2: The differences in the scattering between the 3 examples is not very clear, is it the number, the length of arrows or both, for example what is uniquely different in the scattering between the first and last examples.

Page7, line 14, 'the used orthorectification' is awkward and unclear wording.

Page 7, line 18-19. It is unclear why the minimum value is better than using a mean or median value.

Figure 7: it would be helpful to have the legend on this figure rather than having to refer back to Fig 4.

Page 18, line 33, 'as partially possible for shallow river sections' unclear.

Page 18, line 34, what resolution is meant by 'very high spatial resolution'? Current Hydro1k has rivers at 1km pixels, would 500m be enough to eliminate the mixed pixel impact on of a river on the GM measurements? Figure 14: Label Right hand Y axis, I think it is samples, but the legend has samples/10,000 which would seem to imply the plot is showing a maximum number of samples of 6 x 10ˆ9, which is confusing.

---

## Author Comment (AC2) · 7 Aug 2016

We would like to thank referee #2 for the valuable comments.

On panarctic scale the method and dataset created currently provides in the first place a means to assess the consistency of maps from conventional sources (soil maps). In order to produce a reliable panarctic-map which fully accounts for peat, a combination with other sources (e.g. NCSCD) is required. To use the current version, areas with high SOC content (e.g. >35 kg/m$^2$ for 100cm) should be masked (replaced with other estimates if available). This version will be therefore made available together with a flag (based on NCSCD) which can be used for that. For future studies, we propose in addition a fusion with other land cover information (especially peatland extent from remotely sensed data to spatially confine the high SOC area better). This requires a

[Figure]

reliable circumpolar peatland map, which is to date not yet available.

Figure 2: Many thanks for pointing this out. The length of the arrows represents indeed the backscatter intensity. A higher number of arrows is used when the signal interaction is more complex, when scattering in different directions, with differing intensity occurs.

Choice of backscatter statistics: A mean value could be used in order to account for noise alone. There are however also other effects that need to be accounted for, especially snow related effects such the formation of ice layers due to rain on snow. They would increase the backscatter, but are not expected to be present in all years at that time of the year. The minimum backscatter value (from several years) is used in order to reduce the probability that they affect the backscatter dataset used for SOC retrieval.

Effect of shallow rivers: There is interaction of microwaves at the water – ice boundary of floating ice. If ice on water bodies is freezing to the ground, the backscatter mechanisms change and the backscatter intensity recorded at the sensor drops significantly. This has an effect on SOC retrieval in case of pixels which include very shallow water which is freezing to the bed already in early winter. At circumpolar scale, a river database at 1km like Hydro1k as suggested by the reviewer could be theoretically used for masking pixels which include river courses. The quality of Hydro1k is however in high latitudes rather low.

Figure 14: It corresponds to samples. They are much higher than in figure 9 since the used datasets have a better spatial resolution (more pixels).

---

## Author Response (AR1)

**Editor comment**

1. Please consider to integrate these detailed explanations in the main manuscript text or supplement (if you think this is more feasible).
2. Can you please also address, the reviewer question regarding the use of the L-band and its correlation to map biomass? The reviewer #1 had stated: "What about using L band? Many studies have shown good correlations of L band with vegetation biomass. If the C band better discerns wet and dry surface, than peatland SOC stocks should be predicted better? See P5L18-21."

Regarding reviewer 2:
1. Please integrate your reply to the wider use of the carbon map in the discussion section.
2. Please integrate the corrections and explanations regarding the figures and corrected text in the manuscript.

**Referee #1**

Authors evaluated the use of microwave backscattering Radar data to predict the SOC stocks of Tundra regions. Authors report that the C-band of SAR data can be used to predict the SOC stocks. Validation results show that their approach under predicts the SOC stocks as it does not capture the dynamics of peatlands and cryoturbated soils. The prediction accuracy was reported to decrease with depth. The results of this study are suitable for publication in Biogeosciences. However, at present form, the manuscript uses many undefined acronyms and the language structure is not reader friendly sometimes. The manuscript will benefit from the proper English editing.

*Arctic soils are covered by thick mass of scrub vegetation (herbaceous vegetation less than 5 m tall) or thick O horizon, therefore its unlikely that the microwave spectra can reach to the mineral soil surface. As the Radar microwaves do not penetrate deeper into the soil profile (P6L3-4), the predictions might be a result of pure correlations. It will be helpful to see more descriptions of the process/mechanisms by which the backscattered spectral data is related to the soil properties such as SOC stocks. How the surface vegetation of Tundra region impacts (helps or complicates) SOC predictions?*

> Added to the discussion: The correlation with winter backscatter is expected to result from a combination of roughness (surface response) and volume scattering within the remains of the vegetation (regarding snow, see below). In order to distinguish the different scattering types, polarimetric SAR data as e.g. used in \citet{ullmann14} would be required. Such data are however not available from ENVISAT ASAR GM. Since winter data are used, only interaction with the remaining woody parts is expected. The contribution from volume scattering from woody vegetation becomes important when stems reach a certain size with respect to the used wavelength. The used training and validation sites include also willow dominated landcover. The obtained results from these locations do not indicate that the chosen approach is not applicable. SOC derived from ASAR GM is close to SOC from high resolution optical data (Figure \ref{figure8}) for willow classes. SOC might be however underestimated in case of thicker stems (more than 5 cm).

*This study used backscatter data of December month. What about the surface snow accumulation impacts on backscattering? I think in December, the spectra won't even reach to the soil surface. So basically it can't distinguish between dry and wet areas, which is the basis to estimate SOC stock (as authors describe) in this approach. What about using the spectra of summer months where may be you can find dry and wet*

*areas at surface?*

Background section:
Old: In order to exclude the influence of spatial variations in soil moisture as well as deep snow cover only acquisitions from December (frozen soil, limited snow cover) are used for this study.
New: Tundra and in general wetland environments are commonly classified based on non-frozen period data when SAR data are employed. There are to date only very few studies which make use of frozen period acquisitions \citep{duguay15,widhalm15}. The advantage for using winter data is that only roughness and volume scattering contributes to the return signal intensity. During summer, there is the influence of liquid water in addition. High C-band backscatter areas are therefore often open wetlands (especially peatlands, e.g. \citet{bartsch09,reschke12}}) but can be also areas with high roughness and/or volume scattering. Locations with higher Soil Organic carbon (SOC) are areas with low roughness (with respect to C-band, 5.6 cm wavelength). They have a smoother surface (with respect to the 5.6 cm wavelength) than drier low carbon sites in the high arctic what leads to the hypothesis that C-band backscatter can be used as proxy for SOC content (Fig. \ref{figure2}).

Interaction of the C-band signal with snow grains needs to be accounted for. There is especially an effect when ice crusts form (Naeimi et al. 2012, Bartsch 2010). Backscatter does increase in such cases. C-band is however less sensitive to snow pack changes than shorter wavelengths (such as e.g. Ku-band, Bartsch 2012). Backscatter can increase during the course of the winter by about one db at some locations (Naeimi et al. 2012). In a case study for Yakutia, no increase of ASAR GM backscatter with increasing SWE (snow water equivalent) could be observed \citep{park11}. In order to account for possible contributions by snow cover \begin{enumerate}
    \item only December data should be used, assuming that there are frozen conditions and snow depth is still limited, and
    \item the minimum from as many as possible years should be calculated in order to have the lowest as possible impact (this also accounts for the GM specific noise) \end{itemize}

Added to methods section:
Summer (Juli and August) data have been processed in addition for the Kytalyk site and mean values derived. This dataset is used to exemplify the advantage of using winter data opposed to summer records.

Added to results section:
No relationship ($R^2$ = 0.03) can be found for the SOC zones at Kytalyk in case of comparison to summer backscatter as soil moisture adds to the backscatter of the wetter (and at the same time higher SOC) sites. This differs for winter data. The higher SOC, the lower is the winter backscatter ($R^2$ = 0.74) since it excludes the soil moisture effect (Figure \ref{figure2b}).

*Authors used a variety of data averaging approaches for the evaluation purpose. While generating prediction accuracy or validation errors of maps, I prefer comparison of modeled results with pedon data as done in Figure 12. Please provide R2 values in figure 12.*

comment: Pedon data represent point locations. Due to the very heterogeneous environment it cannot be expected that they are representative for 1km areas (GM resolution). They are therefore only of limited applicability for validation. The zonal maps which have been made based on high resolution optical satellite data are used for the calibration and validation instead. These maps have been quality checked at

all the sites. We have nevertheless decided to show a comparison to the pedon data, since the soil type information (which plays a role in the applicability of the approach, as shown in the comparison with the NCSCD data) is not preserved in the zonal maps. The line in Figure 12 therefore represents only the reference and is not a fitted function.

*Authors calculated RMSE values to demonstrate the prediction accuracy of generated maps (P9L10-12). Please compare the RMSE values with the standard deviation of SOC pedon data at both depths (see Mishra and Riley, 2012). This will provide the predictive ability of the adopted approach.*

Comment: The suggested RPD ratio is not widely used. Its usage is even questioned in the community: B. Minasny: Why calculating RPD is redundant. Pedometron No. 33, August 2013.

Added in the discussion:

Mishra and Riley (2012) obtained an RMSE of 17.8 kg m-2 for the active layer in Alaska using environmental parameters including topography and temperature. Active layer ranges from 14 to 93 cm according to Mishra and Riley (2014). This RMSE is in the same order of magnitude as for the C-band approach (RMSE of 17.24 kg m-2 over 100 cm).

An increase of SOC with mean annual air temperature until a certain threshold followed by an inverted relationship similar as found for the comparison of ASAR GM derived SOC and the length of unfrozen period has been reported by Mishra and Riley (2012) for Alaska.

*What about using L band? Many studies have shown good correlations of L band with vegetation biomass. If the C band better discerns wet and dry surface, than peatland SOC stocks should be predicted better? See P5L18-21.*

Comment: This question is based on the misunderstanding that the underlying parameter of the presented C-band approach is soil moisture. A comparison of summer and winter backscatter has been added to clarify the approach. See also reply to the second question. Mapping vegetation biomass with L-band is applicable for forests (stems need to have a certain size) and thus not relevant for tundra. The following edits to the text have been made:

Old P5L18-21 and following: Regions with soils conditions close to saturation near the surface can be therefore identified using C-band SAR data. This has been demonstrated applicable for peatland detection at high latitudes (Reschke et al., 2012).

New: Regions with soil conditions close to saturation near the surface can be therefore identified using SAR data. This has been demonstrated applicable for peatland detection at high latitudes with C-band (Reschke et al., 2012).

Added to discussion: L-Band (approximately 23cm wavelength) is expected to better penetrate to the ground in the tundra-taiga transition zone. It may in general give better indications of soil moisture during the summer season. The interaction with the surface material (roughness and volume scattering) is however expected to be much lower than with C-band with respect to the tundra surface characteristics (see Fig. 2). The sensitivity to the relevant surface features which are used as proxy for SOC is expected to be lower at L-band.

*Methods: Move the "background" section to Introduction*

The background section focuses on technical aspects of microwave remote sensing methods. The directly following sections build on it. We therefore belief that it should be left for the flow of the manuscript and not separated by the data description section from the methodology.

**Referee #2**

General comments

*The authors present a study that demonstrates the potential use of C-band SAR to determine SOC content in Northern treeless arctic regions. The inferred surface roughness from the SAR data provides an observable metric that may be correlated to measured SOC values from pedon and upscaling land cover based studies. In general the method provides reliable results and at a higher spatial resolution thant the existing NCSCD dataset. The work is innovative and appears to have great promise at better resolving SOC content across permafrost areas.*

*Weaknesses: Where does it go from here? Do the authors believe that it can produce a reliable panarctic map or given the limitations with carbon rich soils do they believe the method should be reserved for areas with traditional landcover-based assessments? If so, will they produce this map? Will it be available?*

Added in the discussion:

In cases where the near surface soil is close to saturation during summer, C-band can be used to distinguish peatlands to some extent \citep{bartsch09,reschke12}. This does however only lead to a yes/no classification. Such maps (or any other appropriate landcover classification) could be used in addition to the presented approach in order to indicate areas where it is expected that SOC is underestimated.

….

On panarctic scale the method and dataset created currently provides in the first place a means to assess the consistency of maps from conventional sources (soil maps). In order to produce a reliable panarctic-map which fully accounts for peat, a combination with other sources (e.g. NCSCD) is required. To use the current version, areas with high SOC content (e.g. >35 kg m\textsuperscript{-2} for 100cm) should be masked (replaced with other estimates if available). For future studies, we propose in addition a fusion with other land cover information (especially peatland extent from remotely sensed data to spatially confine the high SOC area better). This requires a reliable circumpolar peatland map, which is to date not yet available.

*Specific comments:*

*Abstract – last sentence: This sentence is not particularly clear. In general a brief explanation/discussion on why unfrozen period and SOC should be closely related would be very helpful. This relationship is presented in results but early in the paper it would helpful to have more detailed context on the why this is important.*

Old: Comparisons to the length of unfrozen period indicates the suitability of this parameter for modelling of the spatial distribution of soil organic carbon storage.

New: Global Monitoring Mode derived SOC increases with unfrozen period length. This indicates the importance of this parameter for modelling of the spatial distribution of soil organic carbon storage.

*Figure 2: The differences in the scattering between the 3 examples is not very clear, is it the number, the length of arrows or both, for example what is uniquely different in the scattering between the first and last examples.*

Inserted in caption: The length of the arrows represents the backscatter intensity

*Page7, line 14, 'the used orthorectification' is awkward and unclear wording.*
Old: used orthorectification tools
New: tools used for orthorectification

*Page 7, line 18-19. It is unclear why the minimum value is better than using a mean or median value.*

Inserted: The mean value could be used in order to account for noise alone. There are however also other effects that need to be accounted for, especially snow related changes such as the formation of ice layers due to rain on snow. They would increase the backscatter, but are not expected to be present in all years at that time of the year. The usage of the minimum backscatter value (from several years) reduces the probability that structure change affects the backscatter dataset used for SOC retrieval.

*Figure 7: it would be helpful to have the legend on this figure rather than having to refer back to Fig 4.*

Legend has been added

*Page 18, line 33, 'as partially possible for shallow river sections' unclear.*

Old: (as partially possible for shallow river sections)

New: This is partially possible for shallow river sections. There is interaction of microwaves at the water – ice boundary of floating ice. If ice on water bodies is freezing to the ground, the backscatter mechanisms change and the backscatter intensity recorded at the sensor drops significantly. This has an effect on SOC retrieval in case of pixels which include very shallow water which is freezing to the bed already in early winter.

*Page 18, line 34, what resolution is meant by 'very high spatial resolution'? Current Hydro1k has rivers at 1km pixels, would 500m be enough to eliminate the mixed pixel impact on of a river on the GM measurements?*

Old: A very high spatial resolution

New: A water surface map with higher spatial resolution than the used SAR data

*Figure 14: Label Right hand Y axis, I think it is samples, but the legend has samples/10,000 which would seem to imply the plot is showing a maximum number of samples of 6 x 10^9, which is confusing.*

The axis has been now labeled with samples/10000 and the legend with 'MODIS samples'

[revised manuscript text omitted]

(0)(